

# Health and ecological risk of heavy metals in agricultural soils related to Tungsten mining in Southern Jiangxi Province, China

Jinhu Lai[1], Yan Ni[2], Jinying Xu[1] and Daishe Wu[1,3]

[1] School of Resources and Environment and Key Laboratory of Poyang Lake Environment and Resource Utilization of Ministry of Education, Nanchang University, Nanchang, China
[2] College of Ecology and Environment, Yuzhang Normal University, Nanchang, China
[3] School of Materials and Chemical Engineering, Pingxiang University, Pingxiang, China

Corresponding authors
Jinying Xu, xujy2020@ncu.edu.cn
Daishe Wu, dswncu@126.com

## ABSTRACT

**Background:** Dayu County, a major tungsten producer in China, experiences severe heavy metal pollution. This study evaluated the pollution status, the accumulation characteristics in paddy rice, and the potential ecological risks of heavy metals in agricultural soils near tungsten mining areas of Dayu County. Furthermore, the impacts of soil properties on the accumulation of heavy metals in soil were explored.
**Methods:** The geo-accumulation index ($I_{geo}$), the contamination factor (CF), and the pollution load index (PLI) were used to evaluate the pollution status of metals (As, Cd, Cu, Cr, Pb, Mo, W, and Zn) in soils. The ecological risk factor (RI) was used to assess the potential ecological risks of heavy metals in soil. The health risks and accumulation of heavy metals in paddy rice were evaluated using the health risk index and the translocation factor (TF), respectively. Pearson's correlation coefficient was used to discuss the influence of soil factors on heavy metal contents in soil.
**Results:** The concentrations of metals exceeded the respective average background values for soils (As: 10.4, Cd: 0.10, Cu: 20.8, Cr: 48.0, Pb: 32.1, Mo: 0.30, W: 4.93, Zn: 69.0, mg/kg). The levels of As, Cd, Mo, and tungsten(W) exceeded the risk screening values for Chinese agricultural soil contamination and the Dutch standard. The mean concentrations of the eight tested heavy metals followed the order FJ-S > QL > FJ-N > HL > CJ-E > CJ-W, with a significant distribution throughout the Zhangjiang River basin. Heavy metals, especially Cd, were enriched in paddy rice. The $I_{geo}$ and CF assessment indicated that the soil was moderately to heavily polluted by Mo, W and Cd, and the PLI assessment indicated the the sites of FJ-S and QL were extremely severely polluted due to the contribution of Cd, Mo and W. The RI results indicated that Cd posed the highest risk near tungsten mining areas. The non-carcinogenic and total carcinogenic risks were above the threshold values (non-carcinogenic risk by HQ > 1, carcinogenic risks by CR > $1 \times 10^{-4}\,a^{-1}$) for As and Cd. Correlation analysis indicated that $K_2O$, $Na_2O$, and CaO are main factors affecting the accumulation and migration of heavy metals in soils and plants. Our findings reveal significant contamination of soils and crops with heavy metals, especially Cd, Mo, and W, near mining areas, highlighting serious health risks. This emphasizes the need for immediate remedial actions and the implementation of stringent environmental policies to safeguard health and the environment.

# INTRODUCTION

Mining is one of the most economically important activities. However, the heavy metal pollution associated with mining is a major concern (*Cheng et al., 2018*; *Fernández-Martínez et al., 2024*; *Hu et al., 2018*; *Liu, Probst & Liao, 2005*; *Pecina et al., 2023*; *Zhou et al., 2024*). Mining and smelting operations release large quantities of heavy metals into groundwater, atmospheric dust, and runoff (*Han, Golev & Edraki, 2021*; *Hui et al., 2021*; *Mokhtari et al., 2018*; *Mugova & Wolkersdorfer, 2022*). Through these migration pathways, heavy metals can accumulate in the soil, posing ecological risks and health risks *via* bioaccumulation and biomagnification (*Hosseinniaee et al., 2023*; *Kumari & Bhattacharya, 2023*). Additionally, agricultural activities such as planting, livestock rearing, and aquaculture have exacerbated the extensive distribution and bioaccumulation of heavy metals in soil (*Jiang et al., 2017*). This is especially the case for agricultural areas neighboring mining sites irrigated with mining sewage, and heavy metals in soil can have persistent and irreversible toxic effects on various organisms (*Chen et al., 2022*; *Wang et al., 2018*). In this context, it is of great significance to gain insight into the heavy metal pollution of soils in close vicinity to mining and agricultural areas and to understand the potential risks.

Soil properties are crucial factors influencing the distribution and environmental behavior of heavy metals. Research indicates that soil pH and organic matter directly impact the speciation of heavy metals, while soil cation exchange capacity (CEC) regulates the adsorption capacity of heavy metals in the soil (*Liang et al., 2019*). Studies further suggest that soil oxides can influence the transfer and accumulation of heavy metals in crops. For instance, the presence of $K_2O$ as a component in fertilizers affects crop growth, thereby influencing the enrichment of heavy metals (*Lai et al., 2023*). Additionally, CaO and $Na_2O$ can alter soil pH (*Lambers & Barrow, 2020*), and $Fe_2O_3$ can reduce the solubility of metals, thus affecting their accumulation in crops (*Xu et al., 2021*). Existing research suggests the influence of soil properties on heavy metal distribution and accumulation is region-specific, necessitating targeted studies that account for regional variances.

Southern Jiangxi, particularly the southern region of Ganzhou, is recognized as one of the world's crucial tungsten mining centers (*Liao & Zou, 2020*). The substantial wastewater, emissions, and residues generated during the extraction of mineral resources introduce tungsten and associated heavy metals into the surrounding farmlands (*Hu, 2012*), causing significant heavy metal pollution in many agricultural soils. For example, existing studies indicate that in the vicinity of the tungsten mining areas in southern Jiangxi, levels of arsenic, cadmium, mercury, and lead in vegetable gardens and rice fields significantly exceed Chinese soil standards (*Chen et al., 2016*; *Zhou et al., 2021*), resulting in extremely high ecological risks. This severely impedes the healthy development of local agriculture. Therefore, it is essential to conduct relevant research, assess heavy metal pollution, accumulation, and potential ecological and health risks in farmlands around

tungsten mines, laying the theoretical foundation for the sustainable development of agriculture in mining areas.

Dayu County, situated in mountainous terrain north of the Zhangjiang River Basin, Southern Jiangxi Province, China, is a representative region with a long history of tungsten mining, dating back to the early 20[th] century. The mining activities have led to metal pollution of the surrounding agricultural soil (*Wang et al., 2015*; *Liu et al., 2017*). For example, *Wang et al. (2015)* studied heavy metal pollution in farmland around the tungsten mine in Dayu County and reported that 67% of the studied area was contaminated by heavy metals. *Liu et al. (2017)* found that the heavy metals (Pb, Mn, Zn, and Cd) near the tailings mainly occurred in bioavailable acid-soluble forms, posing high ecological risks. However, previous studies mainly focused on evaluating pollution in soil near mining areas or tailing ponds. Up to now, few studies have comprehensively analyzed the pollution and the ecological and health risks of metals in agricultural soils affected by mining activities in Dayu County (*Wang et al., 2015*; *Zheng et al., 2020*), which is of great significance to the health of agricultural development and human health. In this context, the present study investigated six sites surrounding the Dayu tungsten mine, with the following aims: (i) to assess metal pollution (As, Cd, Cr, Cu, Mo, Pb, W, and Zn) in soil along the Zhangjiang River Basin; (ii) to evaluate ecological and health risks posed by heavy metals in paddy rice and soil near the tungsten mining area; (iii) to explore the influence of soil properties on the heavy metals. The results provide a basis for science-based strategies for the remediation and protection of soils in the tungsten mining region.

## MATERIALS AND METHODS

### Study area

Dayu County is located in the southwest of Jiangxi Province, China (E114°–144.44°, N25.15°–25.37°) (Fig. 1). This region contains the highest concentration of tungsten deposits globally, with considerable tungsten-tin mineral occurrences (*Zhang et al., 2021*). The climate is a typical subtropical humid monsoon with distinct four seasons and abundant rainfall. The annual average temperature is between 19.1 °C and 20.8 °C, and the annual average precipitation is 1,318.9 mm. The topography rises in the northern, western, and southern parts of the county, whereas the central and eastern sections form a hilly basin surrounded by mountains on three sides that opens eastward. The Zhangjiang River crosses the county from west to east. The tungsten mines are distributed across the northern mountainous terrain, and farmland occupies the eastern plain. Long-term mineral development has become an important source of heavy metal contamination in this region.

### Soil sampling and chemical analysis

Soil samples were taken at a depth of 0–20 cm at six selected areas in four towns (Fujiang, Huanglong, Qinglong, and Chijiang) along the Zhangjiang River (Fig. 1) on sloping land, sloping plain transition land, and plain land. These sits were chosen considering their agree

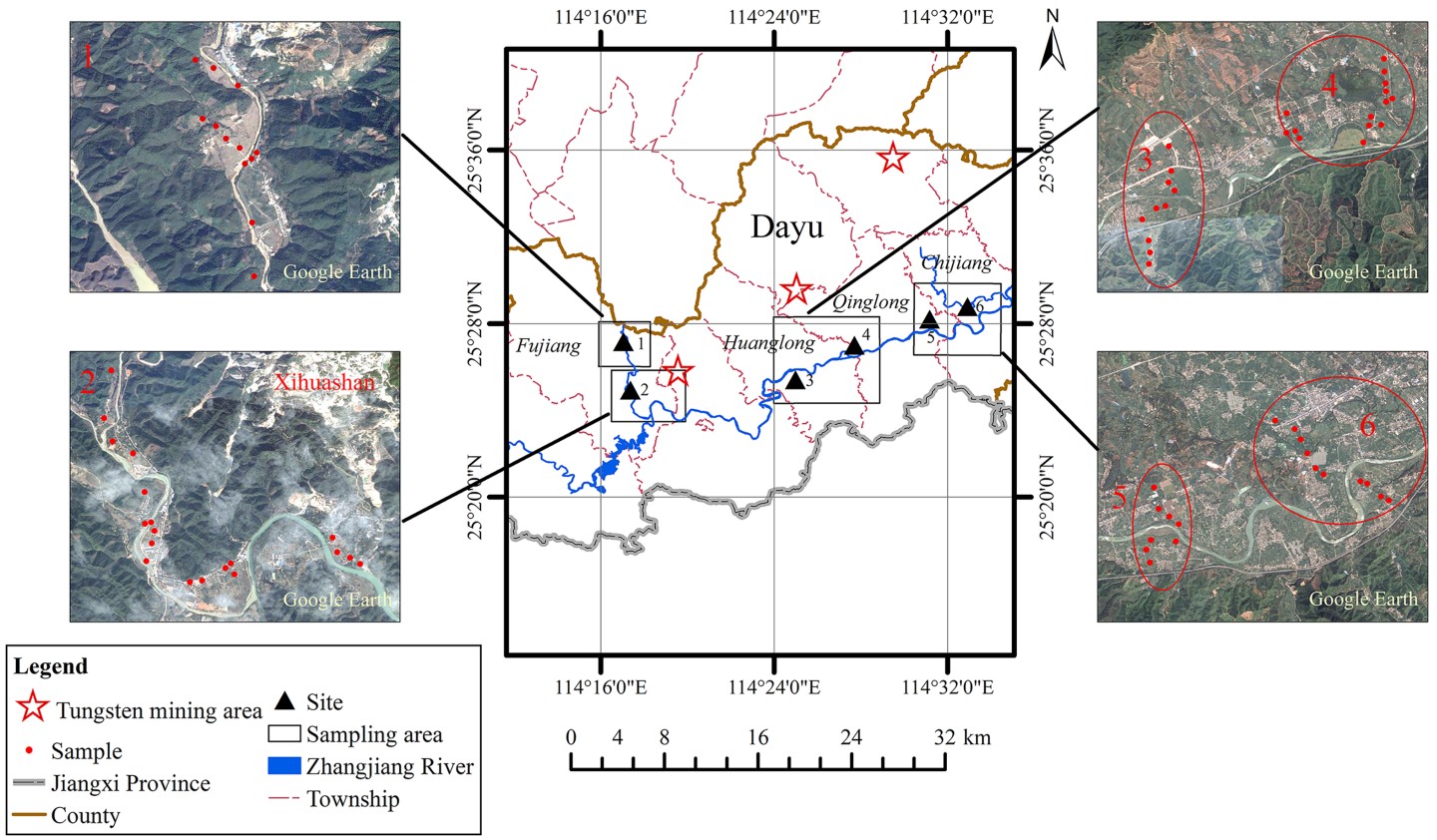

**Figure 1  The study area and sampling sites of soils and paddy rice (site 1: FJ-N, site 2: FJ-S, site 3: HL, site 4: QL, site 5: CJ-W, site 6: CJ-E).** The little red points are samples of soil. The black triangles are sampling sits in study regin. The red five-pointed stars are tungsten mining aeras. The river is represented by blue curve. Other curves represent administrative areas on the map. Site1: FJ-N, Site 2: FJ-S, Site 3: HL, Site 4: QL, Site 5: CJ-W, Site 6: CJ-E; satellite image credit: Google Earth.          

influenced by the tungsten mines: Near Qinglong has a large-scale tungsten mine called Piaotang Tungsten Mine, Fujiang has an important historical tungsten mining area in Xihua Mountain, Huanglong locates no tungsten mine and Chijiang is an important rice planting area far from tungsten mining areas. The soil types in the study area are classified as red soil, yellow soil, and alluvial soil. The six sampling sites were termed FJ-N, FJ-S, HL, QL, CJ-W, and CJ-E; FJ-N and FJ-S are sloping land with artificial terraced fields located along the Fujiang River, a main tributary of the Zhangjiang River, located 4 and 2 km, respectively, from historical tungsten mining areas (Xihuashan, Fig. 1), respectively, HL is a control site far away from mining areas, QL is close to active northern mining areas, and CJ-W and CJ-E are plain sites with traditional paddy rice cultivation. To obtain representative samples, at each site, four to six subsamples were obtained within a 10 × 10-m grid at the center and at the four corners of an ideal square and combined to obtain one composite sample (weight: 1 kg) per site, resulting in a total of 72 soil samples and 26 corresponding rice grain samples. We removed stones, plant debris, and other residues from the soil samples. Due to the influence of terrain, the distance between sampling plots was 50–200 m. Soil samples were obtained using wooden shovels and

stored in polyethylene bags. In the laboratory, all samples were air-dried at room temperature, crushed, homogenized, and sieved through a 100-mesh nylon mesh prior to further analysis. Rice grains were placed in a clean, sunny, and ventilated room on a clean wooden board and frequently turned to prevent mold growth. After drying, grain samples were weighed, sieved through a 100-mesh nylon mesh, and stored in polyethylene bags before digestion.

The concentrations of metals (As, Cd, Cu, Cr, Pb, Mo, W, and Zn) in soils were analyzed using an inductively coupled plasma-mass spectrometer (ICP-MS, iCAP Q; Thermo Fisher Scientific, Waltham, MA, USA). Prior to analysis, the soil and rice samples were digested by a $H_2SO_4$-$HNO_3$-HF mixture using a Microwave Digestion System (Multiwave PRO, Anton Paar, Graz, Austria) (*Li et al., 2006*). A blank control, duplicate samples, and standard reference materials (standard sample of lateritic, GSS-5) were employed to ensure data quality. The concentrations of heavy metals were computed as the average values of duplicate samples. The precision and accuracy of the method is expressed in standard deviation and relative error from GSS-5 ($n = 12$), respectively. The standard deviation (SD %) of As, Cd, Cr, Cu, Mo, Pb, W, Zn is 2.47, 0.02, 1.77, 2.07, 0.16, 2.37, 0.25, 2.60, resepectively. The relative error (RE%) of As, Cd, Cr, Cu, Mo, Pb, W, Zn is 1.12, 2.56, 0.99, −0.26, 2.79, 0.11, −0.55, 1.15, respectively.

In the soil, pH, total organic carbon (TOC), cation exchange capacity (CEC), and metal oxides ($Fe_2O_3$, CaO, $Na_2O$, and $K_2O$) were also measured. The soil pH was measured using a pH meter in a 1:2.5 soil: water mixture, and the soil TOC was determined *via* a Torch TOC combustion analyzer (multi N/C 2100; Analytik Jena, Jena, Germany). The CEC was determined as described elsewhere (*Chen & Huang, 2021*), and the levels of $Fe_2O_3$, CaO, $Na_2O$, and $K_2O$ were measured *via* X-ray fluorescence spectrometry (XRF, EDX-8000; Shimadzu, Kyoto, Japan).

## Assessment of soil heavy metal contamination

(1) Geo-accumulation ($I_{geo}$) index

The geo-accumulation (Igeo) index was employed to assess the pollution degree of heavy metals in soil (*Muller, 1969*), according to Eq. (1):

$$I_{geo} = \log 2 \left[ \frac{C_i}{1.5 B_i} \right], \tag{1}$$

where $C_i$ is the mean concentration of the element in the examined soil, and $B_i$ is the geochemical background value in the soil of Jiangxi Province (As: 10.4, Cd: 0.10, Cu: 20.8, Cr: 48.0, Pb: 32.1, Mo: 0.30, W: 4.93, Zn: 69.0, mg/kg) (*China National Environmental Monitoring Centre (CNEMC), 1990*). The soil pollution levels can be partitioned into seven classes based on the values of $I_{geo}$: Class 0 (unpolluted; $I_{geo} \leq 0$), Class 1 (unpolluted to moderately polluted; $0 < I_{geo} \leq 1$), Class 2 (moderately polluted; $1 < I_{geo} \leq 2$), Class 3 (moderately to heavily polluted; $2 < I_{geo} \leq 3$), Class 4 (heavily polluted; $3 < I_{geo} \leq 4$), Class 5 (heavily to extremely polluted; $4 < I_{geo} \leq 5$), and Class 6 (extremely polluted; $I_{geo} > 5$) (*Loska et al., 1997*).

(2) Contamination factor (CF)

The level of soil contamination is expressed in terms of a contamination factor (CF), calculated following Eq. (2):

$$CF = \frac{C_i}{B_i} \tag{2}$$

where Ci is the mean concentration of the element in the examined soil, and Bi is the background value of metals in the soil of Jiangxi Province (As: 10.4, Cd: 0.10, Cu: 20.8, Cr: 48.0, Pb:32.1, Mo: 0.30, W: 4.93, Zn: 69.0, mg/kg) (*China National Environmental Monitoring Centre (CNEMC), 1990*). The contamination factor CF < 1 refers to low contamination, $1 \le CF < 3$ indicates moderate contamination, $3 \le CF \le 6$ indicates considerable contamination, and CF > 6 indicates very high contamination.

(3) Pollution load index (PLIs)

The overall metal load in soils from each site was computed using the pollution load index (PLIs) (*Tomlinson et al., 1980*) according to Eq. (3):

$$PLIs = (CF_1 \times CF_2 \times CF_3 \times \ldots CF_n)^{\frac{1}{n}} \tag{3}$$

According to Tomlinson, $CF_n$ is the contamination factor of the metal n in the sample. The PLIs provide simple but comparative means for assessing site quality, where a value of PLIs < 1 denotes low contamination, 1 < PLIs < 2 denotes moderate contamination, 2 < PLIs < 3 denotes heavy pollution, and PLIs > 3 denotes extremely heavy pollution.

## Ecological risk assessment regarding metals in soils

The ecological risk factor ($E_r^i$) is a measurement of the potential ecological risk (RI) of a given metal and defined as Eq. (4):

$$E_r^i = T_r^i \left( \frac{C_r^i}{B_i} \right), \tag{4}$$

where $B_i$ is the background value of metals in soil in Jiangxi Province (As:10.4, Cd:0.10, Cu:20.8, Cr:48.0, Pb:32.1, Mo:0.30, W:4.93, Zn:69.0, mg/kg) (*China National Environmental Monitoring Centre (CNEMC), 1990*), $C_r^i$ is the heavy metal content in the soil, $T_r^i$ is the toxic-response factor, and the $T_r^i$ values of Cr, Cu, Zn, As, Cd, Pb, and W are 2, 5, 1, 10, 30, 5, and 2, respectively (*Zheng et al., 2020*). *Hakanson (1980)* introduced the potential ecological risk index (RI) as a method to quantitatively evaluate the risk of soil heavy metal pollution. The RI represents the integrated potential ecological risk, calculated as the sum of individual risk factors ($E_r^i$) for each metal contaminant (*Hakanson, 1980*), and is defined as follows Eq. (5):

$$RI = \sum E_r^i \tag{5}$$

According to *Hakanson (1980)*, $E_r^i$ and RI represent potential ecological risks through different levels (Table 1).

**Table 1 The categories of $E_r^i$ and RI values.**

| $E_r^i$ | | RI | |
|---|---|---|---|
| $E_r^i < 40$ | Low risk | $RI < 150$ | Low risk |
| $40 \leq E_r^i < 80$ | Moderate risk | $150 \leq RI < 300$ | Moderate risk |
| $80 \leq E_r^i < 160$ | Considerable risk | $300 \leq RI < 600$ | Considerable risk |
| $160 \leq E_r^i < 320$ | High risk | $RI \geq 600$ | Very high risk |
| $E_r^i \geq 320$ | Very high risk | | |

Note:
$E_r^i$ and RI represents potential ecological risks through different levels.

## Health risk assessment of heavy metals in paddy rice

As Dayu County is a major rice production area in China, with rice being a staple food for a large part of the population, heavy metal pollution must be considered. The health risk assessment model developed by the United States Environmental Protection Agency (USEPA) was used to evaluate the carcinogenic and non-carcinogenic risks of heavy metals (*United States Environmental Protection Agency, 2011*), using Eqs. (6)–(10) of the average daily dose (ADD$_i$), hazard quotient (HQ) and total hazard quotient index (HI), cancer risk (CR) and total cancer risk (TCR):

$$ADD_i = \frac{C_i \times IR \times EF \times ED}{BW \times AT} \tag{6}$$

$$HQ_i = \sum \frac{ADD_i}{R_f D} \tag{7}$$

$$HI = \sum HQ_i \tag{8}$$

$$CR_i = \sum ADD_i \times SF_i \tag{9}$$

$$TCR = \sum CR_i, \tag{10}$$

where HQ$_i$ and HI are the non-carcinogenic health risks of single and combined metals, respectively, CR and TCR are the carcinogenic health risks of single and combined metals, respectively (*Hui et al., 2021*), ADD$_i$ refers to the daily intake of heavy metal i in paddy rice (mg/kg/day), $C_i$ is the mean value of metal i in rice (mg/kg), EF is the exposure frequency (365 days/year), ED is the exposure duration (30 years), IR indicates the daily intake amount of rice (kg/day), which was 410.13 g/day according to *Ministry of Environmental Protection of China (2013)*, BW is the body weight (60.6 kg), AT refers to the average exposure time (365 × ED days for non-carcinogenic risk; 365 × 70 days for carcinogenic risk), $R_f D$ is the corresponding reference dose of the metals (As: 0.0003, Cd: 0.001, Cr: 0.003, Cu: 0.04, Mo: 0.005, Pb: 0.004, and Zn: 0.3) (mg/kg/day), SF is the slope factor of the metals (As: 1.5, Cd: 6.1, Cr: 0.5, and Pb: 0.0085) (mg/kg/day).

The translocation factor (TF) was used to assess the bioaccumulation abilities of the heavy metals in rice, as described in Eq. (11):

$$\mathrm{TF} = \mathrm{C_{rice}}/\mathrm{C_{soil}} \qquad\qquad (11)$$

## Statistical analysis

The differences in the heavy metal contents in soils from different sampling sites were determined *via* one-way analysis of variance (ANOVA). A Shapiro–Wilk test and Leven's test were performed to ensure the normality and homogeneity of the data prior to ANOVA ($p < 0.05$ or $p < 0.01$). Pearson's correlation analysis was applied to evaluate the influences of soil factors on the heavy metals in the soils. Statistical analyses were conducted in SPSS 26.0 (IBM, Armonk, NY, USA) and Origin 2021b (OriginLab Corporation, Northampton, MA, USA).

# RESULTS AND DISCUSSION

## Concentrations and spatial distribution of heavy metals in soil

Figure 2 and Table S1 show the concentrations and basic statistics of the heavy metals in the tested topsoil samples. The mean concentrations of As, Cd, Cu, Cr, Pb, Mo, W, and Zn exceeded their respective average background values (ABV) of Jiangxi Province soil (*China National Environmental Monitoring Centre (CNEMC), 1990*), accounting for 87.5%, 98.6%, 90.7%, 86.1%, 76.4%, 100%, 97.2%, and 20% in the samples, respectively. The levels of Mo and W surpassed the background values by 34.1 and 19.3 times, respectively, and the Cd concentration exceeded the background level by 6.23 times. The mean values of As, Cu, Cr, Pb, and Zn were 1.33–3.21 times higher than the background values. The coefficient of variation (CV) represents the spatial dispersion and degree of variation of heavy metal concentrations, and the larger its value, the greater the impact of external factors on the pollution level (*Pan et al., 2016*). The CV values for Cd and Mo indicated strong spatial variation (CV > 100%), likely due to anthropogenic activities (*Zhang, 2020*). The levels of Cu, Cr and Zn showed a moderate spatial variation (21% < CV < 50%), and for As, Pb, and W, we observed a high variability (50% < CV < 100%). The risk screening values for agricultural soil contamination are 30, 0.3, 50, 150, 250, and 200 mg kg$^{-1}$ for As, Cd, Cu, Cr, Pb, and Zn, respectively (*Ministry of Ecology and Environment of China (MEEC), 2018*). For Mo and W, the screening values of 3 mg kg$^{-1}$ refer to the Dutch standard (*Dutch Ministry of Housing, Spatial Planning and the Environment (VROM), 2000*). In the present study, the mean As and Cd concentrations were 1.11 and 2.07 times, respectively, their corresponding MEEC limits. However, compared to the Dutch standard, the As and Cu levels indicated mild contamination, whereas the levels for Mo and W suggested strong pollution. In this study, As, Cd, Mo, and W were the major pollutants, similar to the findings for residential aeras near a Russian Russia tungsten mine (*Timofeev, Kosheleva & Kasimov, 2018*). Compared with the findings for a northern tungsten mine in China (*Wu, Wang & Chen, 2020*), in this study, the contents of heavy metal were much lower, except for Mo and Cr. The heavy metal concentrations reported for a southern tungsten mining area were not significantly different to those found in the present study, which could be explained by the soil properties of both sites (*Guo et al., 2017*; *Zheng et al., 2020*). The significant lower

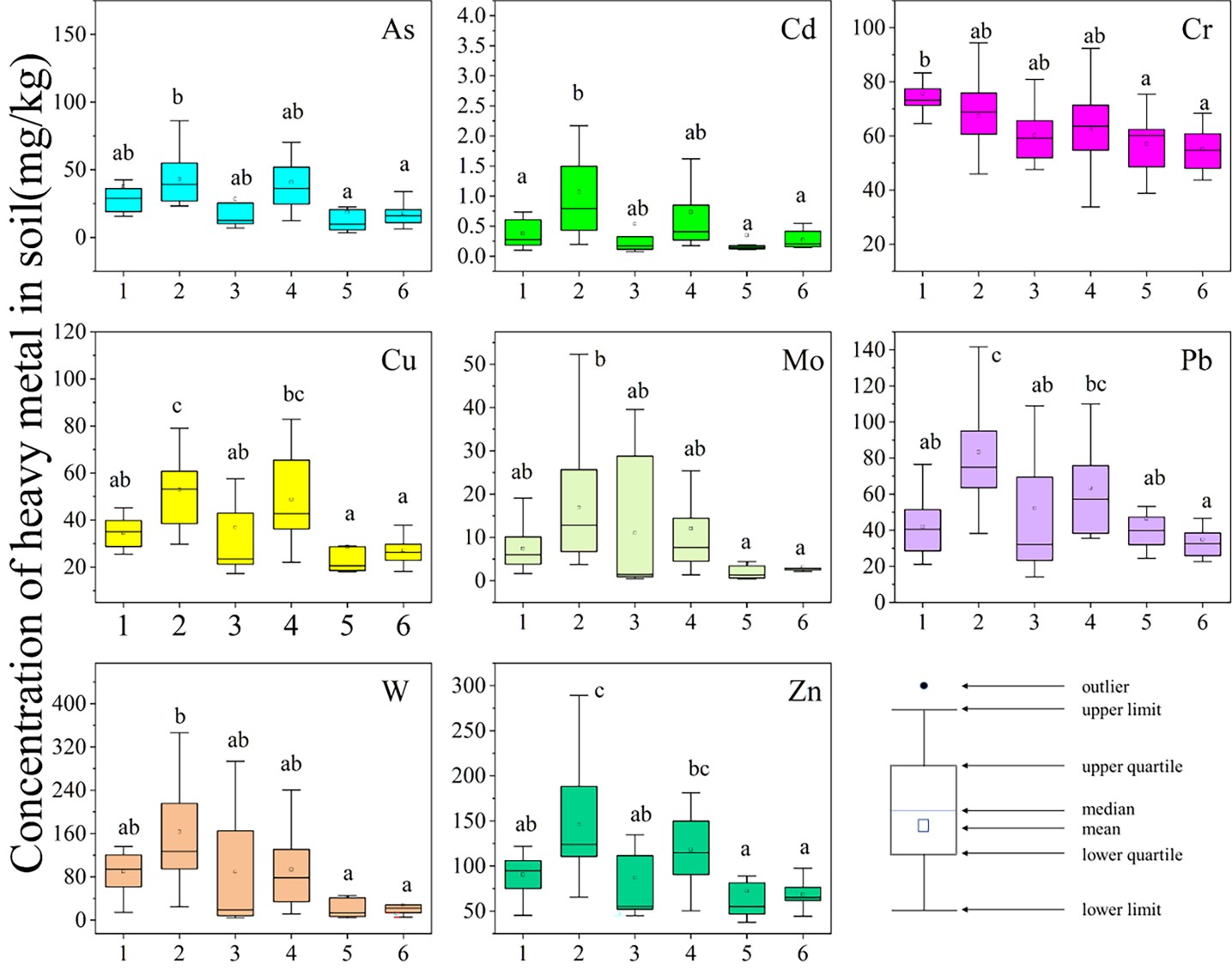

**Figure 2 Concentration of heavy metal in soil (mg/kg).** The X-axis represents the sampling sites. The Y-axis represents the concentration of heavy metals. Box charts show the distribution of data. Different lowercase letters indicate the significance of concentration of heavy metals' differences among sites.

concentration of metals in the study area compared with the northern tungten mining area may be resulted in the distinciton of mining methods, climate types of the two mining areas. Heavy metal pollution in mining areas in southern China is mostly concentrated in river valleys. Due to large rainfall and long duration, heavy metal migration is fast, which is not conducive to accumulation in the soil (*Chen, Fan & Long, 2024*). In the north, it is semi-arid with strong winds, and the dust coverage from open-pit mining is wide and not easily lost, resulting in the accumulation of heavy metals in the soil for a long time (*Mao, Ma & Zou, 2016*).

The mean concentrations of the eight tested heavy metals followed the order FJ-S > QL > FJ-N > HL > CJ-E > CJ-W, with a significant distribution throughout the Zhangjiang

**Table 2 Concentration of components of soil properties.**

| Site | pH | TOC (%) | K$_2$O (%) | Na$_2$O (%) | CaO (%) | Fe$_2$O$_3$ (%) | CEC (mol/kg) |
|------|-----|---------|-----------|------------|---------|----------------|--------------|
| FJ-N | 5.22[ab] ± 0.41 | 2.34[a] ± 0.51 | 1.63[a] ± 0.18 | 0.10[a] ± 0.02 | 0.16[a] ± 0.06 | 4.10[c] ± 0.50 | 0.062[a] ± 0.015 |
| FJ-S | 5.35[b] ± 0.41 | 2.62[a] ± 1.21 | 2.50[b] ± 0.25 | 0.28[b] ± 0.18 | 0.24[bc] ± 0.04 | 3.49[b] ± 0.64 | 0.062[a] ± 0.017 |
| HL | 4.93[a] ± 0.54 | 2.18[a] ± 0.27 | 2.12[ab] ± 0.67 | 0.22[ab] ± 0.08 | 0.20[ab] ± 0.05 | 3.10[ab] ± 0.64 | 0.057[a] ± 0.016 |
| QL | 5.33[ab] ± 0.52 | 1.83[a] ± 0.47 | 2.43[b] ± 0.72 | 0.25[ab] ± 0.11 | 0.26[c] ± 0.05 | 3.48[b] ± 0.91 | 0.065[a] ± 0.018 |
| CJ-W | 5.11[ab] ± 0.40 | 1.85[a] ± 0.44 | 2.54[b] ± 0.58 | 0.21[ab] ± 0.09 | 0.21[abc] ± 0.06 | 2.55[a] ± 0.38 | 0.068[a] ± 0.027 |
| CJ-E | 5.20[ab] ± 0.40 | 2.05[a] ± 0.87 | 2.14[ab] ± 0.59 | 0.21[ab] ± 0.09 | 0.24[abc] ± 0.04 | 2.88[a] ± 0.52 | 0.060[a] ± 0.023 |

Notes:
Different lowercase letters indicate the significance of soil factors' differences among sites.
Compared to concentration of components of Soil properties in sampling sites.

River basin (*Zheng et al., 2020*). The spatial distribution of heavy metal concentrations is shown in Figs. S1 to S8. This indicates that the degree of pollution in the sampling areas was significantly impacted by the distance from the tungsten mining area. Sites FJ-S and FJ-N were only 2 and 4 km, respectively, away from the abandoned mining area (Xihuashan) (*Wang et al., 2009*), with a history of 100 years of mining, resulting in long-term metal accumulation. Site QL was closer to the current mining area Piaotang (*Han, Zhu & Wang, 2019*), with more active metal accumulation. As a control point in the valley on the southern bank of the Zhangjiang River, HL was not only far from historical tungsten mining areas but also was not impacted by the operating mines. Therefore, the concentrations of heavy metals were close to those detected in traditional agricultural areas.

## Correlation between heavy metals and soil properties

Table 2 shows the soil properties for the different sampling sites. Of all soil samples, 73.7% were weakly acidic. The pH values ranged from 3.74 to 6.67, with an average of 5.24 ± 0.46. The average pH value was similar to the background value determined for Ganzhou area (*Chen et al., 2019*; *Wang et al., 2015*, *2019*). The highest and lowest mean pH values were observed for FJ-N (5.37) and HL (4.93), respectively. This variation can be attributed to differences in resident population and human activities, which can notably influence the soil properties (*Shen et al., 2017*; *Vega et al., 2004*).

The concentration of total organic carbon (TOC) in soil samples from site FJ-S was highest, with a mean value of 2.50%. In comparison, soil samples from sites QL, CJ-W, and CJ-E sites showed lower TOC levels, with mean values of 1.83%, 1.85%, and 2.05% respectively. The fractionation and composition of TOC is of great importance when determining the adsorption capacity of soils. The higher TOC content at site FJ-S suggests a greater capacity to adsorb pollutants, likely due to higher inputs of organic matter, such as vegetation litter, expanding the carbon pool. The lower TOC levels in the other sampling sites indicate a more limited organic matter accumulation, constraining the adsorptive potential of the contaminants (*Xiao et al., 2017*). However, at site QL, with high heavy metal concentrations, an increased level of human activities could result in a low TOC level, which was the reason for the bioavailable metal supplementation *via* sewage effluent or mineral fertilizers (*Nacke et al., 2013*).

**Table 3 The Pearson's correlation between soil metals and soil properties.**

|  | As | Cd | Cr | Cu | Mo | Pb | W | Zn |
|---|---|---|---|---|---|---|---|---|
| pH | 0.100 | 0.110 | −0.098 | 0.198 | 0.289* | 0.163 | 0.130 | 0.142 |
| TOC | −0.068 | 0.202 | 0.556** | 0.063 | −0.165 | 0.042 | −0.066 | 0.351** |
| $K_2O$ | 0.386** | 0.504** | −0.326** | 0.520** | 0.583** | 0.681** | 0.490** | 0.483** |
| $Na_2O$ | 0.372** | −0.467** | −0.536** | 0.498** | 0.718** | 0.589** | 0.492** | 0.410** |
| CaO | −0.001 | 0.199 | −0.112 | 0.318** | 0.292* | 0.289* | 0.120 | 0.324** |
| $Fe_2O_3$ | 0.046 | −0.078 | 0.281* | 0.091 | 0.053 | 0.021 | 0.128 | 0.087 |
| CEC | −0.043 | 0.015 | 0.511** | 0.087 | −0.136 | −0.052 | −0.142 | 0.149 |

**Notes:**
\* Correlation is significant at the 0.05 level (2-tailed).
\*\* Correlation is significant at the 0.01 level (2-tailed).

The concentrations of alkali metal oxides ($K_2O$ and $Na_2O$) were below the levels of the continental crustal abundances in China (*Li et al., 2014*). The content of $K_2O$ was significantly higher at FJ-S, QL, and CJ-W (mean value: 2.54%, 2.43%, and 2.54%, respectively), whereas that of $Na_2O$ was significantly higher at FJ-S (mean value: 0.33%). The CaO concentration was higher at sites FJ-S, QL and CJ-E (mean values: 0.25%, 0.26%, and 0.24%, respectively), but this heavy metal was strongly positively correlated with pH (Table S2). At site FJ-N, levels of the above four metallic oxides were lowest. Sites CJ-W and CJ-E had the lowest $Fe_2O_3$ concentration (mean values: 2.55% and 2.88%, respectively).

Based on the Pearson's correlation coefficients (Table 3), Mo was significantly positively correlated with pH ($p < 0.05$), whereas Cr and Zn were highly significantly positively correlated with TOC ($p < 0.01$). The Cr was highly significantly positively correlated with CEC ($p < 0.01$) and significantly positively correlated to $Fe_2O_3$ ($p < 0.05$). These results indicate that higher pH values can facilitate the accumulation of Mo in soil, whereas Cr and Zn are mainly accumulated in the TOC. Consequently, rapid organic degradation leads to rapid Zn cycling (*Liu & Han, 2021*). The Cr levels slightly fluctuated, and Cr was positively correlated with $Fe_2O_3$ and TOC, indicating that it was derived from natural sources. The heavy metals As, Cd, Cu, Mo, Pb, W, and Zn were highly significantly positively correlated with $K_2O$ and $Na_2O$ ($p < 0.01$), whereas CaO was significantly positively correlated with Mo and Pb ($p < 0.05$) and highly significantly positively correlated with Cu and Zn ($p < 0.01$). As seen in Table S5, CaO was strongly positively correlated with pH and CEC, thereby affecting the bioavailability of heavy metals such as Mo, as a result of mine wastewater treatment (*Chen, Zhu & Huang, 2007*; *Meng et al., 2021*). This leads us to infer that $K_2O$ and $Na_2O$ may be principal factors in the accumulation and migration of heavy metals, such as As, Cd, Cu, Mo, Pb, W, and Zn, in soil, whereas CEC and TOC play an important role in Cr accumulation. According to previous studies, $K_2O$, $Na_2O$, and CaO are mainly derived from human activities. Whilst $K_2O$ accumulation is the result of the use of inorganic and organic fertilizers (*Lambers & Barrow, 2020*), $Na_2O$ and CaO are the main agents in the treatment of tungsten ore mining

**Table 4 Heavy metal contamination factors (CFs) and pollution load indexes (PLIs) for metal in soil.**

| Site | Contamination factors (CFs) | | | | | | | | PLIs |
|------|------|-------|------|------|------|------|------|-------|------|
|      | As   | Cd    | Cu   | Zn   | Pb   | Cr   | Mo   | W     |      |
| FJ-N | 3.61 | 3.78  | 1.66 | 1.31 | 1.31 | 1.58 | 12.4 | 17.52 | 3.28 |
| FJ-S | 4.13 | 10.68 | 2.54 | 2.11 | 2.60 | 1.40 | 28.2 | 31.99 | 5.45 |
| HL   | 2.73 | 5.37  | 1.77 | 1.26 | 1.63 | 1.26 | 18.31| 17.49 | 3.48 |
| QL   | 3.93 | 7.33  | 2.34 | 1.71 | 1.97 | 1.30 | 20.1 | 18.36 | 4.26 |
| CJ-W | 1.78 | 3.48  | 1.38 | 1.05 | 1.44 | 1.19 | 6.74 | 8.49  | 2.33 |
| CJ-E | 1.69 | 2.71  | 1.30 | 0.99 | 1.08 | 1.15 | 5.21 | 5.45  | 1.95 |

**Note:**
The data indicates the degree of contamination and pollution level of metals.

wastewater (*Zhang et al., 2012*). This suggests that long-term tungsten ore smelting and agricultural fertilizing have a significant impact on the soil heavy metal content, which has also been reported in previous studies (*Hui et al., 2021*; *Zhang et al., 2017*).

## Pollution and risk assessment of heavy metals in soil

The geo-accumulation index ($I_{geo}$) has been widely used to assess the pollution status of metals in sediments and soils (*Sawe, Shilla & Machiwa, 2019*; *Timofeev et al., 2020*; *Zahra et al., 2014*). In the present study, the mean values of $I_{geo}$ in soil followed the order Mo > W > Cd > As > Pb > Cr > Zn (Table S3). Regarding the pollution classes, Mo and W (with mean values of 2.59 and 2.51, respectively) pollution was moderate to heavy, whereas Cd pollution was moderate (class 3), and for As and Cu, the levels indicated no to moderate pollution. The mean contamination factors (CFs) (*Rastegari Mehr et al., 2017*) for As, Cd, Cu, Cr, Pb, Mo, W, and Zn were 2.98, 5.56, 1.83, 1.31, 1.67, 15.16, 16.55, and 1.41, respectively (Table 4). The contamination factors for Cu, Cr, Pb, and Zn indicated moderate contamination (*Inengite, Abasi & Walter, 2015*) and were significantly lower compared to those for the other tested metals; this leads us to infer that they were derived from natural sources rather than human activities. Although As pollution was moderate, it was close to being considerable. The highest contamination factors were observed for Mo and W, indicating very high contamination (mean values:15.16 and 16.55, respectively), which was due to the long-term tungsten mining in this area (*Wu, 1993*). In contrast, Cd pollution was moderate to high. The main heavy metals found across all sampling sites were Mo, W, and Cd. As their levels greatly exceeded the ABV, they need to be taken into consideration regarding environmental pollution (*Shi et al., 2022*).

Figure 3 shows the $I_{geo}$ spatial distribution of the tested heavy metals. Soil $I_{geo}$ values for As, Cu, Cd, and Zn below 0 were found in sites CJ-W and CJ-E, indicating that agricultural soils were largely unaffected by these metals. For Mo and W, the Igeo values were higher than 2 and lower than 4 in sites FJ-N, FJ-S, HL, and QL, classifying these sites as moderately to heavily polluted and heavily polluted, respectively. For As, moderate pollution was observed, whereas for Cu, Pb, and Zn, the sites were categorized as unpolluted to moderately polluted. Additionally, Cd pollution was moderate to heavy at

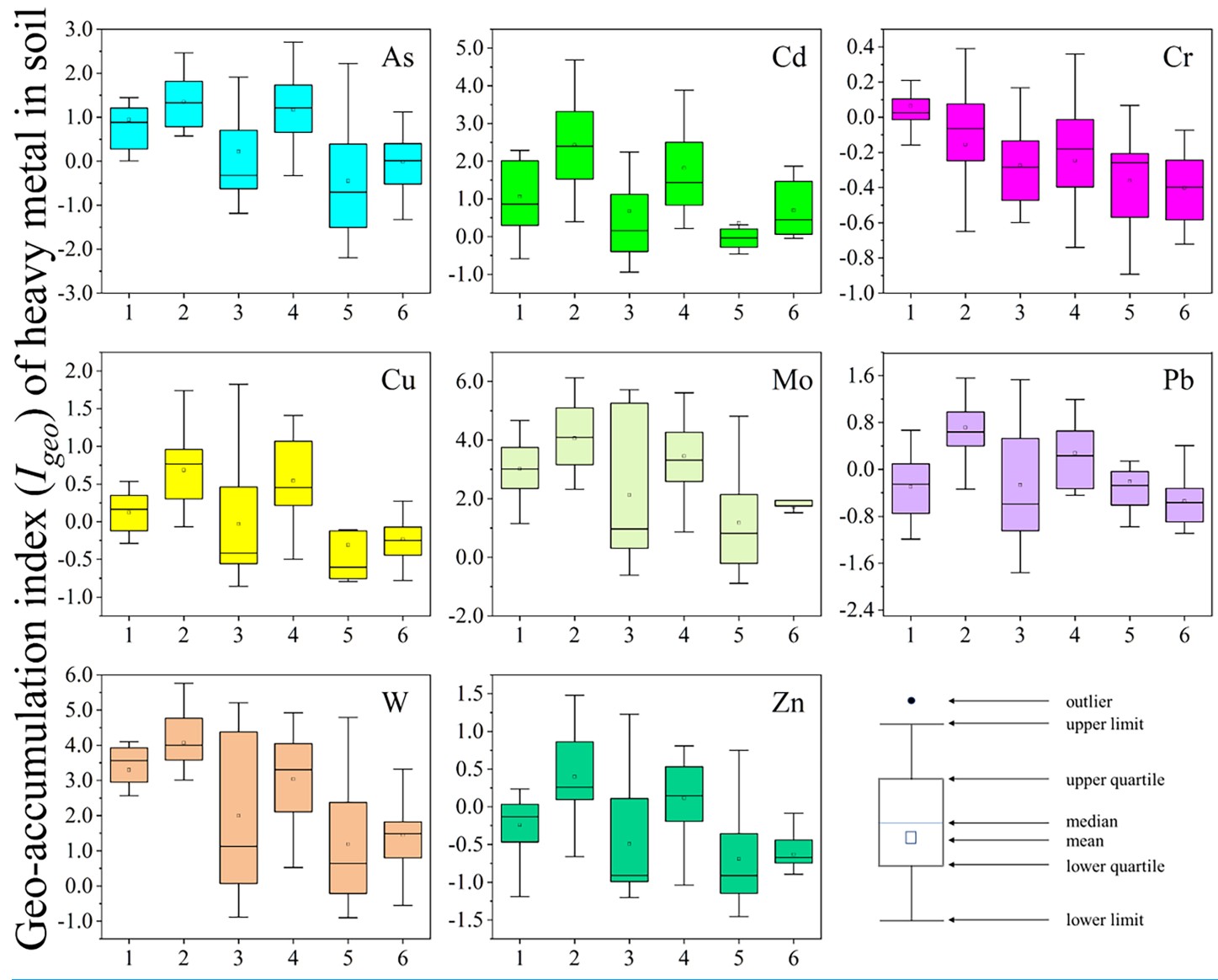

**Figure 3 Geo-accumulation index ($I_{geo}$) of heavy metal in soil in sampling sites (site 1: FJ-N, site 2: FJ-S, site 3: HL, site 4: QL, site 5: CJ-W, site 6: CJ-E).** The X-axis represents the sampling sites. The Y-axis represents the $I_{geo}$ values of heavy metals. Box charts show the distribution of data.

sites FJ-S and QL. Based on the $I_{geo}$ findings, sites FJ-S and QL received varying amounts of heavy metals, except Cr. The PLI findings indicated extremely heavy pollution (PLIs > 3) at sites FJ-S and QL, mostly with Cd, Mo, and W (Table 4). Sites CJ-W and CJ-E exhibited low pollution levels compared to the other sites. The heavy metal accumulation period of site FJ-S was prolonged, and the tailing pond located in the valley was prone to leakage, resulting in the pollution of both surface water and groundwater (*Reutova et al., 2022*). Site QL served as a busy transportation hub for tungsten ore, and the accumulation of heavy metals was worsened by irrigation with sewage (*Lu, Zheng & Lai, 1997*).

**Table 5 Ecological risk of heavy metal in soil.**

| Site | $E_r^i$ | | | | | | | RI |
|------|------|------|------|------|------|------|------|------|
|      | As | Cd | Cu | Cr | Pb | W | Zn | |
| FJ-N | 36.09 | 113 | 8.30 | 3.15 | 6.55 | 35.04 | 1.31 | 204 |
| FJ-S | 41.32 | 320 | 12.72 | 2.81 | 12.98 | 63.98 | 2.11 | 456 |
| HL | 27.28 | 161 | 8.86 | 2.51 | 8.13 | 34.99 | 1.26 | 244 |
| QL | 39.25 | 220 | 11.72 | 2.60 | 9.87 | 36.72 | 1.71 | 322 |
| CJ-W | 17.80 | 104 | 6.88 | 2.38 | 7.20 | 16.99 | 1.05 | 157 |
| CJ-E | 16.93 | 81.30 | 6.50 | 2.30 | 5.42 | 10.91 | 0.99 | 124 |

Note:
The data indicates the ecological risk level of metal in different site.

**Table 6 Concentration of heavy metals in paddy rice.**

| Sites | As ($p < 0.01$) | Cd ($p > 0.05$) | Cr ($p < 0.05$) | Cu ($p > 0.05$) | Mo ($p < 0.05$) | Pb ($p < 0.05$) | Zn ($p < 0.05$) |
|-------|------|------|------|------|------|------|------|
| FJ-N | 0.18[c] ± 0.07 | 0.36[ab] ± 0.01 | 0.08[a] ± 0.001 | 4.31[a] ± 0.88 | 0.005[a] ± 0.002 | 0.05[ab] ± 0.03 | 20.61[bc] ± 3.46 |
| FJ-S | 0.15[bc] ± 0.01 | 1.18[b] ± 0.18 | 0.08[a] ± 0.01 | 5.58[b] ± 0.36 | 0.009[b] ± 0.005 | 0.08[b] ± 0.02 | 21.37[c] ± 1.36 |
| HL | 0.09[a] ± 0.02 | 0.53[ab] ± 0.75 | 0.08[a] ± 0.01 | 4.57[ab] ± 0.68 | 0.005[a] ± 0.002 | 0.04[a] ± 0.01 | 15.55[a] ± 1.53 |
| QL | 0.11[ab] ± 0.03 | 0.83[ab] ± 0.64 | 0.08[a] ± 0.01 | 4.69[ab] ± 0.84 | 0.004[a] ± 0.003 | 0.04[a] ± 0.01 | 17.46[ab] ± 2.29 |
| CJ-W | 0.07[a] ± 0.03 | 0.11[a] ± 0.09 | 0.10[ab] ± 0.01 | 3.86[a] ± 0.68 | 0.001[a] ± 0.0004 | 0.03[a] ± 0.01 | 15.48[a] ± 1.79 |
| CJ-E | 0.11[ab] ± 0.02 | 0.47[b] ± 0.44 | 0.11[b] ± 0.02 | 4.17[a] ± 0.98 | 0.005[a] ± 0.002 | 0.06[ab] ± 0.03 | 18.19[abc] ± 3.45 |

Notes:
The data indicates concentration of paddy rice at different sampling sites.
Different lowercase letters indicates the significant difference of heavy metal concentrations among sites at $p < 0.05$ or $p < 0.01$.

Table 5 shows the ecological risk posed by heavy metals. The ecological risk factor ($E_r^i$) values for Cr, Pb and Zn were below 40 in all sampling aeras. However, for As, the $E_r^i$ values were close to 40 in sites FJ-S and QL (mean: 41.32 and 39.25, respectively), indicating a moderate risk. The $E_r^i$ values of W (mean: 35.04, 63.98, 34.99 and 36.72) had reached or cloesd to a moderate risk in sites FJ-N, FJ-S, HL and QL near the Tungsten mining area. For Cd, the risk degree ranged from considerable risk to very high risk, indicating serious contamination and human health risks. The RI values indicated a considerable risk at FJ-S (value: 456) and a low risk at CJ-E (value: 124), whereas for the other sites, a moderate risk was detected. Risk analysis revealed Cd as the primary contaminant of concern, with sites FJ-S and QL exhibiting the highest risk quotients and contamination levels. Regarding other heavy metals such as As and W, some sites were moderately polluted. The overall ecological risk pattern highlights Cd contamination, especially at sites FJ-S and QL, requiring targeted risk management.

### Concentrations and transfer of metals in paddy rice

Compared to the concentrations of heavy metals for paddy rice (Table 6) (As, $p < 0.01$; Cr, Pb, Mo, Zn, $p < 0.05$; Cd, Cu, $p > 0.05$), the mean values of Cd, Cu, Mo, Pb, and Zn were

the highest in FJ-S, whereas the As concentration was highest in FJ-N. In CJ-W, the heavy metal concentrations for paddy rice were the lowest. The total concentration of heavy metals in paddy rice followed the order FJ-S > FJ-N > QL > CJ-E > HL > CJ-W. For each metal of accumulation in paddy rice followed the order Zn > Cu > Cd > As > Cr > Pb > Mo; the Cr concentration was largely stable. Based on these results, large amounts of heavy metals had accumulated in the farmland next to the historical mining area, which further exacerbated metal accumulation in rice. Active mining areas also release large amounts of heavy metals into the soil. As heavy metals are affected by rainfall and enter irrigation water, they accumulate in rice plants (*Wu, Wang & Chen, 2020*). Based on the TF values, there were differences in the concentration accumulation values (Table S4). Site QL showed the highest TF values for As, Cd, Cu, Mo, and Zn, whereas for Pb and Cr, the highest values were found for CJ-E. For all tested metals, the values were below 1, except for Cd, indicating that paddy rice had a poor ability to accumulate metals in soil. The TF values for Cd were above 2 in FJ-S, HL, and CJ-E and above 5 in QL, indicating serious accumulation. This results is coincident with the study of *Liu et al. (2023)* which indicates that the paddy rice has high TF and BAF values (mean:0.99 and 0.91, respectively) of Cd. *Yang et al. (2021)* reached the opposite conclusion when studying the enrichment of Cd in rice in southern karst geological areas. The high calcium carbonate content in the soil of the karst geological area, which may impede the accumulation of Cd by rice, may explain their different result.

Soil properties and composition play crucial roles in metal accumulation (*Chen et al., 2016*; *Tunc & Sahin, 2017*). In this study, Pearson correlation analysis revealed a significant association of $Na_2O$ and $K_2O$ with the concentration and transfer of specific metals such as Cd. However, this contradicts the expected positive relationship between $Na_2O$, $K_2O$, and Cd (Table S5), which suggests that $Na_2O$ and $K_2O$ might hinder metal accumulation in paddy rice by precipitating metals in soils. Previous research has indicated that sodium and potassium are essential nutrients for plant growth, suggesting that they may induce metal accumulation in plants by promoting plant growth (*Shrivastav et al., 2020*; *Xu et al., 2020*). This could potentially explain the positive influence of $K_2O$ and $Na_2O$ on Cd accumulation observed in this study. Furthermore, $Na_2O$ and $K_2O$ might also enhance Cd accumulation in paddy rice by altering the structure of microorganisms (*Yang et al., 2020*; *Wang et al., 2021*; *Zheng et al., 2021*). Nevertheless, this study did not assess indicators related to rice growth and soil microorganisms. Our future research will continue to investigate the impact of $K_2O$ and $Na_2O$ on Cd accumulation, considering rice growth and soil microorganisms. Additionally, we aim to further explore the primary influencing factors of heavy metal enrichment in rice in mining areas.

## Assessment of the health risks posed by heavy metals in paddy rice

Figures 4, 5 and Table S6 show the results of the health risk assessment. The average daily dose (ADD) values for non-carcinogenic risk were highest for Cd, Cu, and Zn at site FJ-S, exceeding 0.001 mg/kg/day. However, As was the only metal with an ADD above the

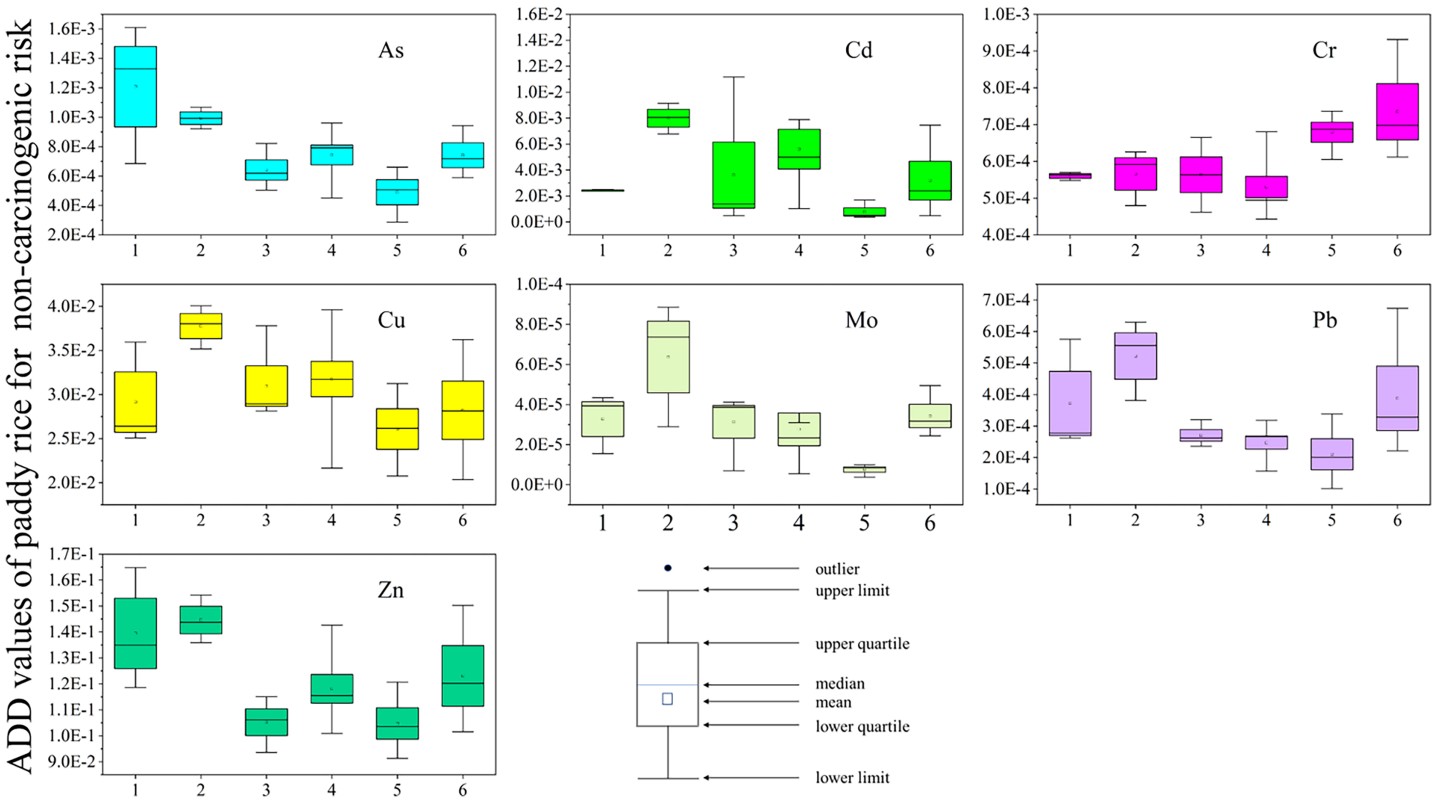

**Figure 4 ADD of heavy metal in paddy rice for non-carcinogenic in sampling areas (site 1: FJ-N, site 2: FJ-S, site 3: HL, site4:QL, site 5: CJ-W, site 6: CJ-E).** The X-axis represents the sampling sites. The Y-axis represents the ADD values of heavy metals. Box charts show the distribution of data.

0.001 mg/kg/day reference limit, specifically at site FJ-N. The hazard quotient (HQ) values for As were above 1 across all study areas, whereas that for Cd was below 1 only at CJ-W. At HQ values above 1, there are potential non-carcinogenic effects on humans (*United States Environmental Protection Agency, 2011*). The order of As HQ was FJ-N > FJ-S > QL > CJ-E > HL > CJ-W (values: 4.03, 3.32, 2.48, 2.47, 2.14, and 1.63, respectively), accounting for 50.88%, 25.40%, 26.05%, 34.69%, 30.01%, and 44.17% of the hazard index (HI), respectively. The levels of Cd HQ followed the order FJ-S > QL > HL > CJ-E > FJ-N > CJ-W (values: 7.99, 5.61, 3.61, 3.18, 2.41, and 0.77), accounting for 61.13%, 58.93%, 50.63%, 44.66%, 30.43%, and 20.87% of the HI, respectively. The HI value exceeded 10 at site FJ-S, indicating health risks and risk of chronic poisoning. The total cancer risk (TCR) was significant across all sites, with As, Cd and Cr cancer risk values(CR) above $1 \times 10^{-4}$ $a^{-1}$ (*Dong et al., 2022*), indicating high carcinogenic risk. Further, As and Cd may represent a considerable potential ecological risk at some sites (FJ-S and QL), based on the pollution characteristics of agricultural soil in mining zones (*Du et al., 2015*). According to the conclusion above, it is crucial to focus on the heavy metal pollution situation at sites FJ-S and QL, reduce the planting of edible crops, and perform soil remediation.

Peer J

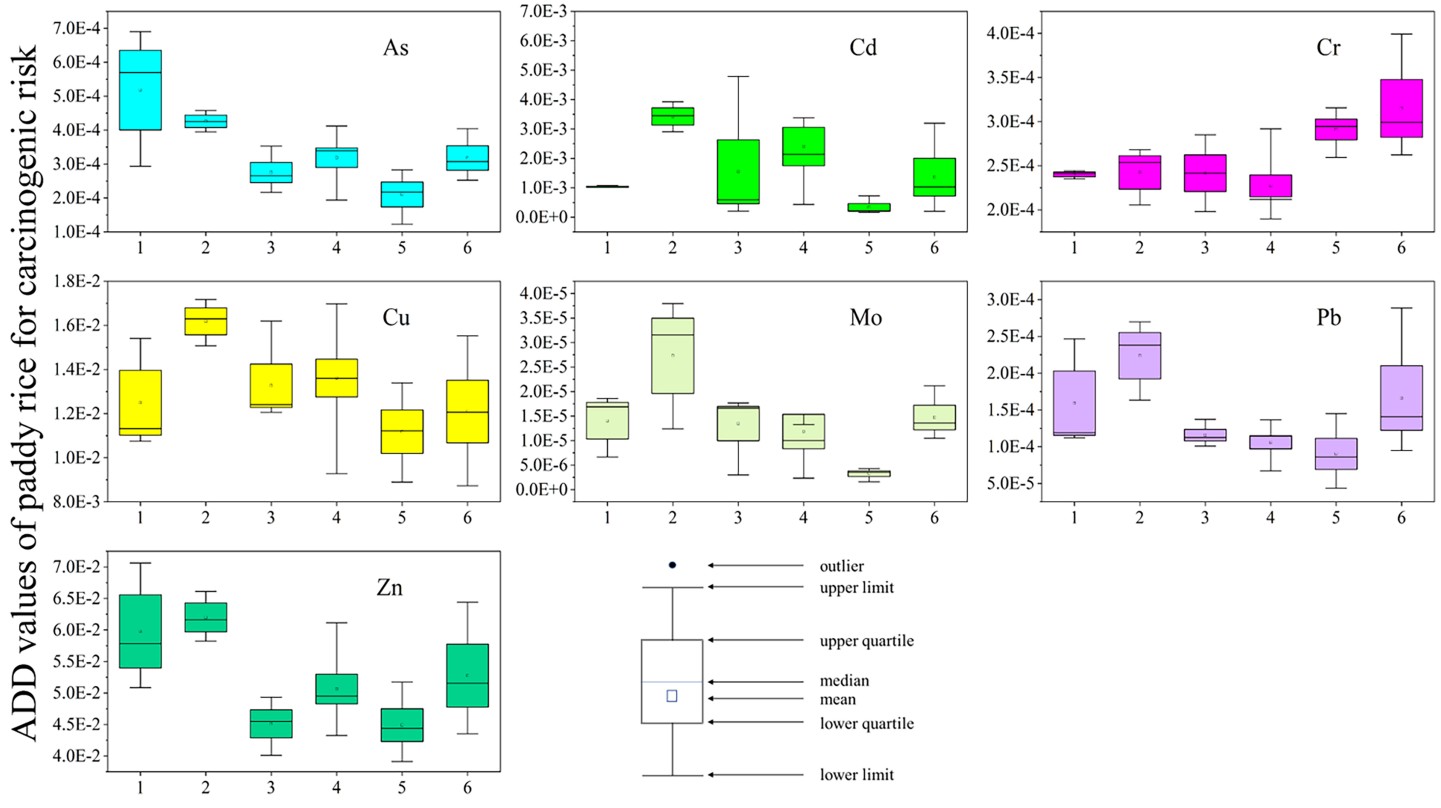

**Figure 5** **ADD of heavy metal in paddy rice for carcinogenic in sampling areas (site 1: FJ-N, site 2: FJ-S, site 3: HL, site 4: QL, site 5: CJ-W, site 6: CJ-E).** The X-axis represents the sampling sites. The Y-axis represents the ADD values of heavy metals. Box charts show the distribution of data.

## CONCLUSIONS

The results of this study indicated that the concentrations of heavy metals were higher at sites near tungsten mines and lower in traditional agricultural areas far from tungsten mines, both for soil and rice samples. In paddy rice, Cd was the most abundant heavy metal. The evaluation of $I_{geo}$ and PLI indicated that As, Cd, Mo and W were the main metals polluted the studied soils. The RI evaluation indicates that Cd and As pose a threat to human health in the study region. Additionally, the correlation analysis indicated that soil $K_2O$, $Na_2O$, and CaO are principal factors affecting the accumulation and migration of heavy metals in soil. This study can provide precious information for managing metal polluted paddy soil near mining area. However, there are limitations for this study. Firstly, the fraction of metals is not considered, which is quite vital for evaluating the pollution and accumulation of metals. Secondly, seasonal change of metal fraction in subtropical area is obvious because of the seasonal rainfall, which is not considered in this study. Therefore, our future research should focus on the pollution and accumulation of metals in soils with considering metal fraction and seasonal rainfall to clarify the potential mechanism.

### Funding

This work was funded by the National Key Research and Development Program of China (2017YF0800900), the Jiangxi Provincial Natural Science Foundation (20224BAB213036) and the Scientific Research Fund of Jiangxi Provincial Education Department (GJJ151327). The funders had no role in study design, data collection and analysis, decision to publish, or preparation of the manuscript.

### Grant Disclosures

The following grant information was disclosed by the authors:
National Key Research and Development Program of China: 2017YF0800900.
Jiangxi Provincial Natural Science Foundation: 20224BAB213036.
Jiangxi Provincial Education Department: GJJ151327.

### Competing Interests

The authors declare that they have no competing interests.

### Author Contributions

- Jinhu Lai conceived and designed the experiments, performed the experiments, analyzed the data, prepared figures and/or tables, authored or reviewed drafts of the article, and approved the final draft.
- Yan Ni performed the experiments, analyzed the data, prepared figures and/or tables, authored or reviewed drafts of the article, and approved the final draft.
- Jinying Xu conceived and designed the experiments, performed the experiments, prepared figures and/or tables, authored or reviewed drafts of the article, and approved the final draft.
- Daishe Wu conceived and designed the experiments, analyzed the data, authored or reviewed drafts of the article, and approved the final draft.

### Data Availability

The raw measurements are available in the Supplemental File.

### Supplemental Information

Supplemental information for this article can be found online at http://dx.doi.org/10.7717/peerj.17200#supplemental-information.

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
