# Peer review of "Health and ecological risk of heavy metals in agricultural soils related to Tungsten mining in Southern Jiangxi Province, China"

_PeerJ, doi:10.7717/peerj.17200_

## Round 0.1 · original submission · Major Revisions

This manuscript requires revisions for language and grammar. Several sections need improved clarity. For instance, the introduction lacks a thorough discussion of the soil and HM relationship. In the Methods and Materials section, additional information is needed in various areas, such as a more comprehensive discussion of the climate in the study area. The data presented in figures and tables is also unclear; certain tabular values require correction. The discussion should follow a more logical flow, and the results should align consistently with the conclusion.

**Language Note:** The review process has identified that the English language must be improved. PeerJ can provide language editing services - please contact us at copyediting@peerj.com for pricing (be sure to provide your manuscript number and title). Alternatively, you should make your own arrangements to improve the language quality and provide details in your response letter. – PeerJ Staff

Reviewer 1 ·

Basic reporting

The manuscript assessed the risk of heavy metals in agricultural soils related to Tungsten mining in Southern Jiangxi Province, China. The heavy metals contents in soil and paddy rice samples were analyzed. The study provide many meaningful data for pollution control in tungsten mining area, but some revision should be done before publication. More comments as follow:
Lines 34: I did not find any information about “Pollution depended on Pollution depended on land use type”.
lines 75-76: the expression was not suitable, many study was conducted to analyzed the pollution and the ecological and health risks of metals in soils affected by mining activities.
Lines 116: In many studies, the digestion procedure for W is significantly different from that of other metals. Make sure your digestion method is feasible.
Lines 134: please provide the background value of heavy metals.
Line 164: In previous studies, the toxic-response factor of W was simply considered to be 1, but in the latest study, a more reasonable method was used to determine that the toxic-response factor of W was 2. The author can consider whether to use this value or not. Details can be found in reference as follow: Li Q, Chen M, Zheng X, et al. Determination of tungsten’s toxicity coefficient for potential ecological risk assessment[J]. Environmental Research Communications, 2023, 5(2): 025003.
lines 244: change Table 1 to Table 2.
Lines 272: change Table 2 to Table 3.
Figure 1 caption: it was red stars in your figure, rather than yellow. Besides, the legend of site and sample were label as different symbols were more suitable.

Experimental design

no

Validity of the findings

no

Additional comments

no

Reviewer 2 ·

Basic reporting

The manuscript is generally clear, but the structure should be improved.

Experimental design

Detailed comments are added.

Validity of the findings

The novelty of this work is not clear.

Additional comments

1. Title-suggest: Health and ecological risk assessment of heavy metals…
2. Line 75-76 The statement is quite weak. Expanding the sampling area is not novel. What is the contribution of this work?
3. Line 114 Why did the authors choose to analyse these metals? Why other toxic metals (e.g., Ni, Mn) not included? Any criteria for the selection.
4. Line 115 How large is the accuracy and precision of this detection method?
5. Line 190 This is incorrect. AT have different values for carcinogenic and non-carcinogenic risk. Please revise and re-estimate the risk values.
6. Line 190 RfD
7. Line 206 Is it better to display the spatial distribution of heavy metals using GIS maps as well as the figure and table here?
8. Line 325 This section should be placed following the section “Concentration and spatial distribution of heavy metals in soil”
9. Line 376-386 The conclusions are too general. What is the novelty of this work comparing to previous studies? What are the implications of the results? What are the future research perspectives?
10. Line 386 lead to heavy metal contamination
11. Figure 2-4 The symbol for the mean value should be added to the box. Also, the distribution of the box (i.e., the representation of different parts) should be described.
12. Table 1 word errors in the caption. Please revise.
13. Table 2 ab bc a b c How large is the significance? The descriptions should be added.

Reviewer 3 ·

Basic reporting

This paper is based on the risk of heavy metal pollution in agricultural soil of tungsten mines in southern Jiangxi Province. Using soil heavy metal data from 6 stations in Dayu County, multiple evaluation methods are used to analyze the status of soil heavy metal pollution. This study uses health risk indices and transport factors to discuss the impact of soil factors on soil heavy metal content, elucidates the correlation between various factors affecting soil pollution, and infers the main factors affecting the accumulation of heavy metals in soil. This is a meaningful research and attempt!

Experimental design

However, there are several issues that need to be noted in the paper
(1) There is no detailed description of the relationship between soil and heavy pollution, and the introduction to the southern part of Jiangxi Province and Dayu County is too repetitive and lengthy, without highlighting the relevant background of tungsten mining agricultural ecological risk assessment;
(2) The article lacks specific introduction on the impact of soil composition on heavy metal pollution;
(3) The author did not provide explanations or brief introductions to the selected area in the study area, including hydrological conditions, climate characteristics, topography, water level, precipitation, and other factors.;
(4) In soil sampling and chemical analysis, the author did not provide a brief explanation of the soil type, specific sampling methods, human activities around the sampling area, and industrial conditions of the samples taken;
(5) The conclusion that soil KO2, NaO2, and CaO are the main factors affecting the accumulation and migration of heavy metals in soil has not been analyzed and reflected in the article;
(6) The results of this article do not consider the chemical reactions between heavy metal ions in natural soil, and the author needs to explain whether there is an impact on the error of the research results.

Validity of the findings

Other issues are as follows:
(1) It is recommended to use concise language to condense the abstract and summary expression to ensure that readers can accurately understand the purpose and methods of the research.
(2) Some references show formatting and font issues, and it is recommended to modify them.
(3) There are issues with some numerical values and font formats in Tables 1, 2, 3, and 5, and it is recommended to adjust them.
(4) The concentration and spatial distribution of heavy metals in soil in the article did not provide a detailed explanation of the differences in the distribution of heavy metals at various points.
(5) The migration of heavy metals in rice is not explained in detail in the article, nor is there a hierarchical analysis of heavy metal changes.
(6) Figure 1 shows that the size and color of the sampling points in the study area are unclear and overlap issues occur, and the serial numbers are not marked. It is recommended to modify them.
(7) Try to unify the color of the box diagram format for each metal in Figures 2, 3, and 4.
(8) The specific steps and methods for handling soil samples are missing in the materials and methods. It is recommended to supplement the formulas or steps for various evaluation methods such as soil accumulation index (Igeo), pollution factor (CF), and pollution load index (PLD), rather than simply introducing their principles.
(9) The evaluation of the health hazards of heavy metals in rice in the article should be placed after the discussion of the correlation between heavy metals and soil properties
(10) In the conclusion section, it is recommended to elaborate on each point and supplement relevant risk management measures for heavy metals.

---

## Round 0.2 · Minor Revisions

Some corrections still need correction as also pointed out by the reviewers. Please carefully address the reviewer comments. In addition, cross-check all the references in the text and also in the full list. Remove the errors from tabular data and ensure that all tables and figures are properly cited in the text.

Reviewer 1 ·

Basic reporting

The author has made sufficient revisions to the article based on the comments of the reviewers, and the article is ready for publication. However, some details should be noted. For example, lines 439 and lines 448 are the same reference but listed twice.

Experimental design

no comment

Validity of the findings

no comment

Additional comments

no comment

Reviewer 2 ·

Basic reporting

For comment 12, there are still word errors in Table 1. like --[endif]--> a

Experimental design

no.

Validity of the findings

For comment 8 on "display the spatial distribution of heavy metals using GIS maps", it is very strange that the authors replied that the study area is so large and the distance among the six sampling sites is long. This is not true since the authors plotted GIS map of Figure 1. Please add the GIS maps on the concentration of heavy metals in the revised manuscript.

Additional comments

no.

---

## Round 0.3 · Minor Revisions

Please see my annotated PDF file for comments.

Reviewer 1 ·

Basic reporting

The author has fully revised the article and it is now ready for publication

Experimental design

none

Validity of the findings

none

Additional comments

none

---

## Round 0.4 · accepted · Accept

The authors have thoroughly addressed all comments provided by the reviewers. I am satisfied with the current version of the manuscript, and it is now ready for publication.